# Gut Microbiota Differentially Mediated by Qingmao Tea and Qingzhuan Tea Alleviated High-Fat-Induced Obesity and Associated Metabolic Disorders: The Impact of Microbial Fermentation

**DOI:** 10.3390/foods11203210

**Published:** 2022-10-14

**Authors:** Lizeng Cheng, Yang Wei, Lurong Xu, Lanlan Peng, Yuanfeng Wang, Xinlin Wei

**Affiliations:** 1School of Agriculture and Biology, Shanghai Jiao Tong University, 800 Dongchuan Road, Shanghai 200240, China; 2College of Life Sciences, Shanghai Normal University, 100 Guilin Road, Shanghai 200234, China

**Keywords:** Qingzhuan tea, microbial fermentation, antiobesity activity, metabolic disorder, gut microbiota, microbiomic analysis

## Abstract

Although dark tea is a unique microbial-fermented tea with a high reputation for having an antiobesity effect, little is known about the effect of microbial fermentation on tea leaves’ antiobesity properties. This study compared the antiobesity effects of microbial-fermented Qingzhuan tea (QZT) and unfermented Qingmao tea (QMT), providing insight into their underlying mechanisms associated with gut microbiota. Our results indicated that the supplementation of QMT extract (QMTe) and QZT extract (QZTe) displayed similar antiobesity effects in high-fat diet (HFD)-fed mice, but the hypolipidemic effect of QZTe was significantly stronger than that of QMTe. The microbiomic analysis indicated that QZTe was more effective than QMTe at regulating HFD-caused gut microbiota dysbiosis. *Akkermansiaceae* and *Bifidobacteriaceae*, which have negative correlations with obesity, were enhanced notably by QZTe, whereas *Faecalibaculum* and *Erysipelotrichaceae*, which are positively correlated with obesity, were decreased dramatically by QMTe and QZTe. A Tax4Fun analysis of QMTe/QZTe-mediated gut microbiota revealed that QMTe supplementation drastically reversed the HFD-induced upregulation of glycolysis and energy metabolism, whereas QZTe supplementation significantly restored the HFD-caused downregulation of pyruvate metabolism. Our findings suggested that microbial fermentation showed a limited effect on tea leaves’ antiobesity, but enhanced their hypolipidemic activity, and QZT could attenuate obesity and associated metabolic disorders by favorably modulating gut microbiota.

## 1. Introduction

The devastating epidemic of obesity, which is a metabolic disease, has become a serious public health problem globally. According to the statistics of the World Health Organization (WHO), the number of adults who suffer from obesity has increased nearly tripled since 1975, reaching as high as 650 million worldwide in 2016. Obesity is highly associated with fat accumulation and chronic inflammation, and predisposes individuals to a series of metabolic disorders including hyperlipidemia, hyperglycemia, and nonalcoholic fatty liver disease (NAFLD) [1]. It even significantly elevates the risk of developing cardiovascular disease and certain types of cancers [2]. Thus, the prevention and therapy of obesity have become major challenges for modern societies.

Dark tea is a unique microbial-fermented tea with a high reputation for having an antiobesity effect and causing metabolic improvements. Several studies in HFD-induced obese rodents have shown that dark-tea intervention for 2 months markedly ameliorated obesity and related metabolic disorders, as evidenced by the reduction in body weight, inhibition of fat accumulation, and improvement of lipid profiles [3,4]. In patients with obesity, dark-tea supplementation for 3 months effectively lowered their body mass index (BMI) and waist–hip ratio (WHR), and alleviated visceral fat accumulation and abdominal obesity [5,6]. Interestingly, the inhibitory effect of dark tea on obesity and associated metabolic disorders was found to be stronger than that of other types of tea. For example, comparative studies using rats treated with various types of tea proved that dark tea was significantly more effective than green tea and black tea in inhibiting weight gain, and more effective than green tea, oolong tea, and black tea in improving hyperlipidemia in rats with metabolic disease [7]. Dark tea reportedly has significantly stronger hepatoprotective activity than green tea, oolong tea, and black tea [8]. Compared with other teas, dark tea is quite different due to its microbial fermentation, in which numerous indigenous microorganisms grow vigorously and dramatically alter the chemical composition of raw tea leaves. Although the antiobesity effect of dark tea has been well documented in rodents and human subjects, it is still unclear whether microbial fermentation has distinctive effects on tea leaves’ antiobesity properties.

Accumulating evidence demonstrated that the antiobesity effect of dark tea is connected with its regulation of gut microbiota. Ye et al. [9] investigated the modulatory effect of dark tea on obesity in HFD-fed mice, and found that its antiobesity effect was associated with its regulation of gut microbiota, such as the restoration of microbial diversity, the increase in beneficial *Bacteroides* and *Akkermansia*, and the decrease in obesity-related *Faecalibaculum* and *Erysipelatoclostridium*. In a report by Liu et al. [10], a dark-tea intervention significantly meliorated obesity and associated metabolic disorders in healthy mice, whereas germ-free mice were resistant to the dark-tea-mediated antiobesity effect and metabolic improvements, indicating that the gut microbiota was required for dark tea to exert its antiobesity effect and metabolic improvements. A fecal transplant trial by Lu et al. [11] indicated that the intervention, as seen in the feces from dark-tea-treated mice, substantially reduced the body weight and alleviated metabolic syndrome in the recipient mice, suggesting that the antiobesity effect of dark tea acted through the gut microbiota. All of these findings suggested that dark tea could significantly modulate gut microbiota dysbiosis, and the gut microbiota mediated by dark tea could improve obesity and associated metabolic disorders. However, the effect of microbial fermentation on the tea leaves’ regulation of gut microbiota and gut-microbiota-mediated improvements in obesity have rarely been reported.

Qingzhuan tea (QZT), mainly produced in Hubei Province of China, is a typical dark tea manufactured through the microbial fermentation of Qingmao tea (QMT). It has always been consumed as an indispensable beverage for nomads, who live in China’s border areas, Mongolia, and Russia with high-fat diet (HFD) as their staple diet, for its health benefits. It is well known that the long-term intake of an HFD predisposes individuals to obesity and associated metabolic disorders, whereas obesity, hyperlipidemia, and NAFLD are less likely to occur in these ethnic groups who co-consume QZT. Coinciding with this observation, Feng et al. [12] recently proved that QZT intervention significantly reduced body weight and fat accumulation and improved serum TC and TG in HFD-fed mice. Our previous study revealed that microbial fermentation dramatically changed the chemical ingredients of QMT and played a key role in forming the special sensory quality of QZT [13]. As a part of our continuing effort in studying QZT, herein, we compared the modulatory effect of microbial-fermented QZT on obesity with that of unfermented QMT in HFD-induced obese mice, highlighting the impact of microbial fermentation. Meanwhile, the alteration of gut microbiota in response to QMT and QZT intervention was deeply investigated by microbiomic analysis, providing new insight into how QMTe/QZTe-mediated gut microbiota alleviate obesity and associated metabolic disorders.

## 2. Materials and Methods

### 2.1. Preparation of QMT and QZT Extracts

The unfermented QMT and microbial-fermented QZT samples were produced by Xinding Biological Technology Co., Ltd. (Yichang, Hubei Province, China) following our previously described method [13], in which they were produced in 2019 in the same production batch. QMT and QZT were milled into fine powder (100 mesh). A total of 200 g of QMT or QZT powder was mixed with 4000 mL of boiled water and extracted in boiling water for 30 min. The extract was centrifuged (Velocity 14R; Dynamica, Livingston, UK) at 5000× *g* for 15 min. The sediment was re-extracted using the same method. The supernatants were pooled and concentrated under vacuum, and finally, lyophilized, producing QMT extract (QMTe) and QZT extract (QZTe). The chemical compositions of QMTe and QZTe were analyzed using our previously reported methods [13]. QMTe contained 33.04 ± 0.56% tea polyphenols, 28.50 ± 0.44% total flavonoids, 20.07 ± 0.22% tea polysaccharides, and 8.30 ± 0.19% theabrownins. QZTe contained 18.64 ± 0.91% tea polyphenols, 18.56 ± 0.36% total flavonoids, 27.06 ± 1.20% tea polysaccharides, and 17.46 ± 0.45% theabrownins.

### 2.2. Animals and Experimental Design

A total of 40 male C57BL/6J mice (3–5 weeks old, body weight 14.70 ± 0.80 g) were purchased from Shanghai SLAC Laboratory Animal Co., Ltd. (Shanghai, China; SCXK<Hu>2017-0005) and housed in the Laboratory Animal Center of Shanghai Jiao Tong University (Shanghai, China) under a regulated barrier system (20–26 °C, 40–70% relative humidity, and 12 h light–dark cycle). Experimental groups and respective treatments are described in Figure 1A. After a 1-week acclimatization with a normal diet (12% fat, 3.616 Kcal/g), all mice were randomly allocated into the control group (*n* = 10 mice) or model group (*n* = 30 mice), and were fed with a normal diet or HFD (28% fat, 5.128 Kcal/g), respectively. After 10 weeks of HFD feeding, the obesity model was successfully established as indicated by the dramatic increase in body weight (BW) (Figure 1B). Mice in the model group were further randomly divided into HFD, QMT, and QZT groups (10 mice per group). The QMT and QZT groups were gavaged with 450 mg/kg BW QMTe and 450 mg/kg BW QZTe, respectively, whereas the HFD group was gavaged with an equal volume of sterile water, once daily, for 10 weeks. All mice had access to food and water ad libitum, and food intake and body weight were recorded weekly. At the end of the experiment, all mice were fasted overnight (12 h), anesthetized with isoflurane for the collection of blood (from the eye socket vein), and sacrificed for the collection of subcutaneous fat, liver, and colonic content. The blood (*n* = 6 samples per group) was centrifuged (4 °C, 3500× *g*, 10 min) and the serum was obtained for follow-up biochemical analysis. The subcutaneous fat and liver (*n* = 6 samples per group) were weighed and soaked in formalin for histopathological observation. The colonic content (*n* = 6 samples per group) was collected in sterilized tubes and immediately frozen in liquid nitrogen for microbiomic analysis.

### 2.3. Biochemical Analysis and Histopathological Observation

The serum levels of triglycerides (TG), total cholesterol (TC), low-density lipoprotein cholesterol (LDL-C), high-density lipoprotein cholesterol (HDL-C), aspartate aminotransferase (AST), alanine aminotransferase (ALT), and alkaline phosphatase (ALP) were measured using commercially available kits (Rayto; Shenzhen, Guangdong Province, China). Histological observation of the subcutaneous fat and liver samples was carried out according to a previous method [10]. Briefly, the formalin-soaked subcutaneous fat and liver were gradient dehydrated, embedded in paraffin, and sliced into 3 μm sections. All sections were stained with hematoxylin and eosin (H&E), and scanned using a DMS-10-Pro digital pathology slide scanner (D·metrix; Suzhou, Jiangsu Province, China).

### 2.4. 16S rRNA Amplicon Sequencing of Gut Microbiota

The genomic DNA of the colonic content samples was extracted using the QIAamp DNA Stool Mini Kit according to the manufacturer’s protocols (Qiagen, Hilden, Germany). The extracted DNA was amplified with barcoded specific primers targeting the V4 hypervariable region of the 16S rRNA gene using the primers 515F 5′-barcode CCTAYGGGRBGCASCAG-3′ and 806R 5′-GGACTACNNGGGTATCTAAT-barcode-3′. The sequencing library was generated using the TruSeq^®^ DNA PCR-Free Sample Preparation Kit (Illumina, San Diego, CA, USA) and was assessed using the Qubit 2.0 Fluorometer (Thermo Scientific, Waltham, MA, USA) and Agilent Bioanalyzer 2100 system. The library was sequenced on an Illumina NovaSeq 6000 platform by Panomix (Suzhou, Jiangsu Province, China). After splicing and aligning, the sequences were clustered into operational taxonomic units (OTUs) using Uparse (version 7.0.1001, http://drive5.com/uparse/ (accessed on 5 September 2022)), with a 97% sequence similarity. The gut microbiota diversity and microbial taxa distribution analyses were analyzed using the QIIME (Quantitative Insights into Microbial Ecology, version 1.9.1, http://qiime.org/scripts/split_libraries_fastq.html (accessed on 5 September 2022)) software. The significantly differential taxa among these groups were identified by the linear discriminant analysis (LDA) effect size (LEfSe) analysis, with the LDA score set at 4.0. The functional profiles of the gut microbiota were analyzed using Tax4Fun (http://tax4fun.gobics.de/ (accessed on 5 September 2022)) based on the 16S Silva Database.

### 2.5. Statistical Analyses

All experimental data were expressed as mean ± standard deviation (SD) and subject to a one-way ANOVA using GraphPad Prism software (version 6.02). The correlation between the most differential taxa and obesity-related phenotypes was analyzed via Spearman’s correlation. The statistical significance was set as follows: not significant (ns), *p* > 0.5; *, *p* ≤ 0.05; **, *p* ≤ 0.01; and ***, *p* ≤ 0.001.

## 3. Results and Discussion

### 3.1. The Effects of QMTe and QZTe on Body Weight Gain, Subcutaneous Fat Accumulation, and Hepatic Lipid Deposition in HFD-Induced Obese Mice

To investigate the impacts of microbial fermentation on tea leaves’ antiobesity properties and metabolic improvements, the HFD-induced obese mice were supplemented with QMTe and QZTe for 10 consecutive weeks. As shown in Figure 1B,C, HFD feeding for 10 weeks significantly increased body weight gain and subcutaneous fat accumulation, which were increased by 40.32% and 138.37%, respectively, compared with the control group. As expected, QZTe supplementation dramatically suppressed body weight gain and subcutaneous fat accumulation in HFD-fed mice. The liver index, which is expressed as the ratio of liver weight to body weight, directly reflects the degree of lipid deposition in the liver. The increase in the liver index induced by the HFD was also markedly inhibited by the QZTe treatment (*p* < 0.001, Figure 1E), suggesting a reduction in hepatic lipid deposition. The protective effect of QZTe on HFD-induced obesity was further verified via the histopathological analyses of subcutaneous fat and liver. As depicted in Figure 2A, HFD feeding drastically facilitated adipocyte hypertrophy, whereas the QZTe treatment strikingly inhibited the adipocyte enlargement in HFD-fed mice. Likewise, the HFD-induced hepatic lipid deposition was alleviated by the QZTe treatment, as evidenced by the attenuation of hepatocellular microvesicular steatosis and inflammatory cell infiltration (Figure 2B). These results indicated that QZTe treatment could significantly suppress body weight gain and subcutaneous fat accumulation, and alleviate hepatic lipid deposition in HFD-fed mice, which coincided with the antiobesity effects of other kinds of dark tea. It has been reported that pu-erh tea dramatically reduced the characteristics of obesity and ameliorated the serum biochemical profile in mice fed an HFD [9]. In the study of Yoo et al. [14], the administration of Fuzhuan tea extract significantly lowered body weight and adipose tissue mass, and improved serum biochemical parameters and hepatic steatosis concomitant in HFD-fed mice.

Despite the chemical compositions varying significantly between QMTe and QZTe, the body weight, subcutaneous fat accumulation, and liver index in the QMT group and QZT group showed no significant difference (Figure 1B,C,E; *p* > 0.05). It was also observed that the attenuation of adipocyte hypertrophy and hepatocellular microvesicular steatosis by the QZTe treatment was similar to that by the QMTe treatment (Figure 2A,B). These results suggested that the antiobesity effect of QZTe was comparable to that of QMTe, and the effect of microbial fermentation on tea leaves’ antiobesity activity was insignificant. In support of this result, raw pu-erh tea and ripened pu-erh tea exhibited similar antiobesity effects in HFD-induced obese mice [9]. Likewise, no marked difference in the prevention of obesity in HFD-fed mice was discovered among green tea, oolong tea, and black tea [15,16]. Although tea polyphenols of QMT were substantially transformed into theabrownins of QZT during the microbial fermentation, both polyphenols and theabrownins are the main bioactive components contributing to the antiobesity effects of tea leaves [17,18]. This may be the reason why microbial fermentation has an insignificant effect on tea leaves’ antiobesity properties. During the 10 weeks of dietary intervention, mice treated with QMTe and QZTe ate an equal amount of food (*p* > 0.05, Figure 1D) but gained less weight (*p* < 0.05, Figure 1B) when compared with mice in the HFD group. Therefore, it is reasonable to hypothesize that the antiobesity effects of QMTe and QZTe were not due to the decrease in food intake, and QMTe and QZTe supplementation may lower the energy efficiency of HFD-fed mice. Consistent with this finding, Gao et al. [19] also reported that the inhibitory effect of pu-erh tea on obesity is mostly related to a reduction in energy efficiency.

### 3.2. QMTe and QZTe Intervention Alleviated Lipid Disorders and Hepatic Damage in HFD-Induced Obese Mice

The serum biochemical parameters TC, TG, LDL-C, and HDL-C reflect the status of lipid absorption and metabolism in HFD-fed mice. As depicted in Figure 1F–I, the levels of serum TC, TG, LDL-C, and HDL-C in the HFD group were elevated 1.20–2.09-fold compared with those in the control group. The high levels of TC and LDL-C are risk factors for coronary heart disease, whereas the high level of HDL-C favorably transports excess cholesterol to the liver for excretion and protects against atherosclerosis and coronary artery disease [20]. The elevated level of HDL-C in the HFD group, which was also observed in the studies of Feng et al. [12] and Liu et al. [10], might be feedback from the large intake in the HFD. It is very encouraging to find that the administration of QZTe remarkably reduced the levels of TC, TG, and LDL-C, and increased the level of HDL-C (*p* < 0.01). Especially, the levels of TG and LDL-C in the QZT group were restored to the levels of the control group (*p* > 0.05). These results indicated that the QZTe intervention potently attenuated lipid disorders in HFD-fed mice, which were in agreement with the lipid-lowering effect of pu-erh tea and Fuzhuan tea [10,11]. QMTe intervention also improved hyperlipidemia in HFD-fed mice, as revealed by the reduction in TC and LDL-C levels (vs. the HFD group, *p* < 0.01). However, QMTe had a limited effect on the serum levels of TG and HDL-C, as identified by the fact that their serum levels were individually higher and lower than those in the QZT group (*p* < 0.05) and showed no significant difference when compared with those in the HFD group (*p* > 0.05). These findings suggested that QMTe also alleviated hyperlipidemia in HFD-fed mice, but its effect was less effective than that of QZTe. Deng et al. [21] obtained a similar result, showing that microbial-fermented pu-erh tea exhibited a better hypolipidemic effect than its unfermented raw material. Kuo et al. [7] also reported that pu-erh tea (a type of microbial-fermented tea) was more effective than green tea (unfermented tea) in inhibiting hyperlipidemia. Compared with QMTe, QZTe contained significantly lower contents of tea polyphenols and total flavonoids, but higher contents of tea polysaccharides and theabrownins. It has been revealed that tea polysaccharides and theabrownins are the major active ingredients contributing to the hypolipidemic property of dark tea. The administration of dark tea polysaccharides for 1–2 months markedly regulated the gene expression of lipid metabolism, reduced the levels of TC, TG, and LDL-C, increased the level of HDL-C, and ameliorated lipid oxidation in rats/mice with hyperlipidemia [22,23]. In HFD-induced hyperlipidemic mice/rats, a theabrownin intervention effectively prevented the increase in serum TC, TG, and LDL-C levels, and inhibited the decrease in the serum HDL-C level [24,25]. The higher contents of polysaccharides and theabrownins in QZTe may account for its stronger hypolipidemic activity in comparison to QMTe.

The liver is the primary site for lipogenesis and lipid metabolism, and in turn, easily suffers from HFD-induced lipid deposition and damage. Serum enzymes, including AST, ALT, and ALP, are regarded as the most sensitive biomarkers of liver damage. It can be seen that the levels of serum AST, ALT, and ALP in the HFD group were all remarkably higher than those in the control group (*p* < 0.001, Figure 1J–L), suggesting liver damage due to HFD feeding. Additionally, the QZTe intervention reverted the levels of AST, ALT, and ALP to the levels of the control group (*p* > 0.05). Despite a slight variation, the levels of AST, ALT, and ALP between the QMT group and QZT group were not statistical different (*p* > 0.05). These results indicated that the effect of microbial fermentation on tea leaves’ hepatoprotective activity was insignificant, and QMTe and QZTe treatment equally suppressed HFD-induced lipotoxicity in the liver.

### 3.3. The Alteration of Microbiota Diversity and Overall Structure in Response to QMTe and QZTe Intervention

To investigate the regulatory effect of QMT and QZT on gut microbiota, the microbial profile of the QMTe/QZTe-treated mice was measured using the 16S rRNA amplicon sequencing. In total, 1,384,413 effective 16S rRNA reads were observed from 24 colonic content samples with an average of 57,684 effective reads per sample, and 18,095 distinct operational taxonomic units (OTUs) were obtained. As reflected in Figure 3A, the species accumulation boxplot of all samples was eventually stable, indicating that the sequencing data covered the vast majority of microbial species and were suitable for further microbiomic analysis. HFD feeding significantly decreased gut microbiota diversity, as indicated by the reduction in the chao1 index (Figure 3B). A reduction in microbiota diversity is one of the main features of defective colonization and has been associated with several metabolic disorders [26]. Expectedly, the decline of gut microbiota diversity caused by an HFD was reversed by QZTe intervention, as reflected by the restoration of the chao1 index. Notably, the chao1 index showed no statistical difference between the QMT group and HFD group, suggesting that QMTe intervention showed a limited effect on gut microbiota diversity in HFD-fed mice.

The principal coordinate analysis (PCoA) was performed to view the effects of QMTe and QZTe intervention on the overall structure of the gut microbiota. Of interest is the observation that the HFD group was quite distinct from the control group, and the QMT group was still closer to the HFD group, whereas the QZT group tended to be closer to the control group (Figure 3C). These results indicated that the HFD and QZTe had profound impacts on the gut microbiota structure, and the gut microbiota dysbiosis induced by the HFD was returned back to a normal status through QZTe intervention. The gut microbiota dysbiosis enhanced the energy harvested from the gut, and induced damage to the intestinal integrity and the release of endotoxin lipopolysaccharides (LPS) from intestinal Gram-negative bacteria into the bloodstream, resulting in obesity and metabolic inflammation [27]. It was reported that pu-erh tea prevented individuals from being obese via the rebalancing of the gut microbiota [11]. The restoration of gut microbiota by the QZTe treatment may contribute to its antiobesity effect in HFD-fed mice. This finding also implied that the modulatory effect of QZTe on gut microbiota dysbiosis was generally stronger than that of QMTe. A similar result has been coded by Ye et al. [9], where ripened pu-erh tea was more effective than raw pu-erh tea at regulating the HFD-induced gut microbiota dysbiosis in obese mice. Taken together, these results demonstrated that QZTe was more effective than QMTe at restoring gut microbiota diversity and regulating gut microbiota dysbiosis in HFD-fed mice.

### 3.4. The Most Differentially Abundant Taxa Enriched by QMTe and QZTe Intervention

The effects of QMTe and QZTe treatment on the abundance of gut microbiota at the phylum and family levels are shown in Figure 3D,E, respectively. Taxonomic profiling indicated that the composition of gut microbiota was dominated by *Bacteroidetes* and *Firmicutes* at the phylum level, contributing to 49.60% and 44.88% of the gut microbiota in the control group, 22.00% and 71.10% in the HFD group, 36.62% and 57.05% in the QMT group, and 11.56% and 71.45% in the QZT group, respectively. The QMTe treatment showed a limited effect on the *Firmicutes*-to-*Bacteroidetes* (F/B) ratio, whereas the QZTe treatment significantly increased the F/B ratio in comparison to the HFD group (*p* < 0.01; Appendix A). In line with this finding, the administration of pu-erh tea substantially upregulated the F/B ratio in diet-induced obese rats [28]. Gu et al. [29] also reported that obesity-resistant mice had a relatively higher F/B ratio in comparison to the obesity-prone mice. The most abundant microbiotas at the family level were *Erysipelotrichaceae* and *Muribaculaceae*, contributing to 26.26% and 42.21% of the gut microbiota in the control group, 56.83% and 13.97% in the HFD group, 42.70% and 26.54% in the QMT group, and 48.07% and 7.95% in the QZT group, respectively. Our data demonstrated that QMTe supplementation significantly increased the relative abundances of *Muribaculaceae*, whereas both QMTe and QZTe supplementation profoundly decreased the relative abundances of *Erysipelotrichaceae* (Appendix A). In accord with this finding, the increase in the relative abundance of *Erysipelotrichaceae* induced by the HFD was also restored by the intervention of Fuzhuan tea and its polysaccharides [10,23]. A bloom of *Erysipelotrichaceae*, which was usually found in diet-induced obese animals and individuals, could enhance the energy extraction of hosts from the diet and shape the host’s obesity phenotypes and lipidemic disorders [30,31]. *Muribaculaceae* was significantly negatively correlated with obesity-related indexes and was regarded as a biomarker of a healthy mouse microbiome [32]. These findings suggested that QMTe and QZTe could attenuate obesity and associated metabolic disorders by mediating certain gut microbiota.

The most differentially abundant taxa in response to QMTe and QZTe intervention was identified by the LEfSe analysis (Figure 4). At the family level, the relative abundance of *Erysipelotrichaceae* significantly increased after HFD feeding, and then decreased with QMTe and QZTe intervention (Figure 5), which coincided with the result of taxonomic profiling. HFD feeding also profoundly decreased the relative abundance of *Clostridiaceae*, *Eubacterium*_*coprostanoligenes*_group, and *Lactobacillaceae* (*p* < 0.01), wherein the decrease in *Lactobacillaceae* was normalized by QZTe intervention. It is worth noting that QMTe had significant enrichment effects on *Bacteroidaceae* and *Lachnospiraceae*, whereas QZTe had significant enrichment effects on *Akkermansiaceae* and *Bifidobacteriaceae*. At the genus level, the marked attenuation of *Lactobacillus* by HFD feeding was restored by QZTe treatment, and the HFD-caused increase in *Faecalibaculum* was reduced by QMTe and QZTe intervention (Figure 5). Meanwhile, QMTe significantly enriched *Bacteroides* and *Lachnospiraceae*_NK4A136_group, and QZTe remarkably enriched *Akkermansia* and *Bifidobacterium*. Particularly, QZTe intervention normalized the HFD-induced increase in *Erysipelatoclostridium*, and both QMTe and QZTe supplementation downregulated the HFD-induced increase in *Blautia*, *Mucispirillum*, and *Romboutsia* to the level of the control group (Appendix A). At the species level, the relative abundance of *Faecalibaculum*_*rodentium* was substantially increased by HFD feeding, and then, reverted by QMTe and QZTe treatment. The relative abundance of *Bacteroides*_*vulgatus* was significantly enhanced by QMTe treatment, whereas QZTe treatment dramatically enhanced the relative abundance of *Akkermansia*_*muciniphila*, *Bifidobacterium*_*pseudolongum*, and *Lactobacillus*_*johnsonii*. Thus, the QMTe intervention preferentially enriched *Bacteroidaceae* (including the subgroups *Bacteroides* and *Bacteroides_vulgatus*) and *Lachnospiraceae* (including the subgroup *Lachnospiraceae*_NK4A136_group), whereas the QZTe intervention preferentially enriched *Akkermansiaceae* (including the subgroups *Akkermansia* and *Akkermansia muciniphila*) and *Bifidobacteriaceae* (including the subgroups *Bifidobacterium* and *Bifidobacterium*_*pseudolongum*) in HFD-fed mice. In support of such findings, the relative abundance of *Bacteroidaceae* and *Lachnospiraceae* was also significantly increased by the treatment of green tea polyphenols in HFD-induced obese mice [33]. The intervention of pu-erh tea, Fuzhuan tea, and their polyphenols was also found to increase the relative proportion of *Akkermansia* (including *Akkermansia muciniphila*) and *Bifidobacteriaceae* in obese mice/rats [11,19,34].

### 3.5. Gut Microbiota Differentially Mediated by QMTe and QZTe Alleviated HFD-Induced Obesity and Associated Metabolic Disorders

Spearman’s correlation analysis was carried out to reveal the gut microbiota contributing to the inhibitory effects of QMTe and QZTe on HFD-induced obesity. The heatmap (Figure 5) displayed the association between the most differentially abundant taxa (altered by HFD, QMTe, and QZTe intervention, shown in Figure 4) and obesity-related phenotypes, including body weight, subcutaneous fat, liver index, TC, TG, LDL-C, HDL-C, ALT, AST, and ALP. It was revealed that the *Lactobacillales*, *Lactobacillaceae*, and *Lactobacillus* were negatively correlated with TC and TG, and *Lactobacillus*_*johnsonii* was negatively related to body weight and TG. In supporting this observation, Wang et al. [35] reported that the relative abundance of *Lactobacillales* was highly negatively linked with TC and LDL-C in HFD-fed mice. Yang et al. [36] also found that *Lactobacillus* could enhance the activity of bile brine hydrolysis enzymes and the content of unbound bile acids, and thus, increase the fecal lipid excretion and reduce lipid accumulation. The enrichment of *Lactobacillales* and *Lactobacillus*_*johnsonii*, and the restoration of *Lactobacillaceae* and *Lactobacillus* by QZTe treatment, may contribute to reducing body weight and improving lipid levels in HFD-fed mice. It was also observed that *Lachnospirales*, *Lachnospiraceae*, and *Lachnospiraceae*_NK4A136_group were negatively related to body weight. This result is consistent with previous studies in which *Lachnospiraceae* was butyric acid-producing bacteria and the administration of *Lachnospiraceae* attenuated obesity, inflammation, and dysbiosis [37,38]. The enrichment of *Lachnospirales*, *Lachnospiraceae*, and *Lachnospiraceae*_NK4A136_group by QMTe treatment might potentially contribute to its antiobesity effect. *Akkermansiaceae*, *Akkermansia*, and *Akkermansia*_*muciniphila* were found to be negatively correlated with body weight and TG, whereas *Bifidobacteriales*, *Bifidobacteriaceae*, *Bifidobacterium*, and *Bifidobacterium*_*pseudolongum* were highly negatively correlated with TG and positively related to HDL-C. Our findings are in accordance with previous studies in which a high relative abundance of *Akkermansia* was associated with a low risk of obesity [39], and the administration of *Akkermansia muciniphila* protected HFD-fed mice against obesity and associated complications by reducing energy efficiency and body weight, promoting the browning of white adipose tissue, and improving lipid metabolism disorder [19,40]. It was also reported that *Bifidobacterium* supplementation reduced the HFD-induced increase in TG and TC via the deconjugation of bile acids, and thus, improved hyperlipidemia in obese mice [41]. The enrichment of *Akkermansiaceae* (including the subgroups *Akkermansia* and *Akkermansia muciniphila*) and *Bifidobacteriaceae* (including the subgroups *Bifidobacterium* and *Bifidobacterium*_*pseudolongum*) by QZTe treatment potentially contributed to its antiobesity effect and metabolic improvement in HFD-fed mice. *Bacteroidota*, *Bacteroidia*, and *Bacteroidales* were negatively associated with body weight and TC, and *Bacteroidaceae*, *Bacteroides*, and *Bacteroides*_*vulgatus* were negatively associated with the increase in body weight, which is in agreement with the previous findings. Ye et al. [9] revealed that *Bacteroides*, which could metabolize carbohydrates and proteins via glycolysis and pentose phosphate pathways, were negatively correlated with obesity and its comorbidities. Zhang et al. [42] also reported that *Bacteroidales* and *Bacteroides* were negatively correlated with weight gain, TC, and TG. The restoration of *Bacteroidota*, *Bacteroidia*, and *Bacteroidales*, and the increase in *Bacteroidaceae*, *Bacteroides*, and *Bacteroides*_*vulgatus* by QMTe intervention may be conducive to its inhibition of obesity. Conversely, *Faecalibaculum* is highly related to weight gain and plays an important role in metabolic disease [43,44]. Consistently, it was observed that *Faecalibaculum* and *Faecalibaculum*_*rodentium* were closely related to subcutaneous fat and TC, whereas *Erysipelotrichales* and *Erysipelotrichaceae* were strongly correlated with TC. The supplementation of QMTe and QZTe significantly reduced the HFD-induced increase in *Faecalibaculum* and *Faecalibaculum*_*rodentium* and restored the HFD-induced increase in *Erysipelotrichales* and *Erysipelotrichaceae* to the normal level, which may be conducive to their mitigation of obesity and lipidemic disorders in HFD-fed mice. These gut microbiota showed significant association with the metabolic parameters and might be the most effective taxa contributing to preventing obesity and associated metabolic disorders.

To further understand how QMTe/QZTe-mediated gut microbiota prevents obesity and associated metabolic disorders, the 16S rRNA gene sequencing data in this experiment were analyzed by Tax4Fun and compared with the microbial database (16S Silva Database) of known metabolic functions. Based on the changes in gut microbiota in each group, it was observed that multiple metabolic functions were significantly altered by HFD feeding, whereas the intervention of QMTe and QZTe restored these changes (Figure 6). For instance, glycolysis and energy metabolism in the HFD group were profoundly increased, whereas the intervention of QMTe drastically reversed these upregulations. The QZTe intervention markedly upregulated the HFD-caused decrease in pyruvate metabolism. Consistently, the pyruvate metabolism in obese men was also substantially enriched after 24 weeks of combined training [45]. Pyruvate is the final product from glucose in the glycolysis pathway, and can be converted into acetyl-CoA and form citrate (the first compound in the TCA cycle) in aerobic metabolism [46]. The pyruvate metabolism upregulated by QZTe intervention may enhance the switching from fat to carbohydrate oxidation, thus promoting energy expenditure and attenuating fat accumulation. The starch and sucrose metabolism was markedly enhanced by HFD feeding, whereas both QMTe and QZTe intervention significantly inhibited this function. It is well known that starch and sucrose metabolism is positively correlated with the regulatory ability of energy metabolism [47]. The inhibition of the starch and sucrose metabolism by QMTe and QZTe supplementation may suppress the energy harvested from the gut and have the ability to reduce obesity. Likewise, the QMTe and QZTe treatment markedly reversed the HFD-induced downregulation of arginine biosynthesis. It was also found that the QMTe and QZTe intervention resulted in a variety of metabolic functions. For example, the intake of QMTe significantly reduced the sphingolipid metabolism, and stimulated valine leucine and isoleucine biosynthesis and pentose and glucuronate interconversions, whereas the QZTe intervention dramatically weakened the glycerolipid metabolism, and enhanced the TCA cycle and tryptophan metabolism in HFD-fed mice. Sphingolipid, which was found to be elevated in obese rodents and human subjects, was among the most pathogenic lipids at the onset of the sequelae associated with excess adiposity [48]. The inhibition of sphingolipid metabolism by QMTe intervention may contribute to its improvement in obesity. Tryptophan metabolism is a key metabolic pathway in the inflammatory infiltration of the liver associated with NAFLD, in which tryptophan effectively mitigated hepatitis via reducing the levels of pro-inflammatory cytokines [49]. The enhancement of tryptophan metabolism by QZTe treatment may alleviate liver damage in HFD-fed mice.

## 4. Conclusions

To the best of our knowledge, this is the first study comparing the antiobesity effects of microbial-fermented QZT and unfermented QMT, providing insight into their underlying mechanisms associated with gut microbiota. Our results demonstrated that QMTe and QZTe displayed similar antiobesity effects in HFD-fed mice, as reflected by the equally reduction in body weight, fat accumulation, and hepatic lipid deposition, but the hypolipidemic effect of QZTe was significantly stronger than that of QMTe. Microbiomic analysis indicated that QZTe was more effective than QMTe at restoring gut microbiota diversity and regulating gut microbiota dysbiosis in HFD-fed mice. Phylotypes of *Akkermansiaceae* and *Bifidobacteriaceae*, which have negative correlations with obesity, were enhanced notably by the intervention of QZTe, whereas *Faecalibaculum* and *Erysipelotrichaceae*, which are positively correlated with obesity, were decreased dramatically by both QMTe and QZTe treatment. Tax4Fun analysis of QMTe/QZTe-mediated gut microbiota revealed that QMTe supplementation drastically reversed the upregulation of glycolysis and energy metabolism, and QZTe supplementation significantly restored the downregulation of pyruvate metabolism, whereas both QMTe and QZTe supplementation markedly reduced the increase in starch and sucrose metabolism in HFD-fed mice. Our findings suggested that microbial fermentation showed a limited effect on tea leaves’ antiobesity, but enhanced their hypolipidemic activity, and QZT could greatly attenuate obesity and associated metabolic disorders by favorably modulating gut microbiota.

## Figures and Tables

**Figure 1 foods-11-03210-f001:**
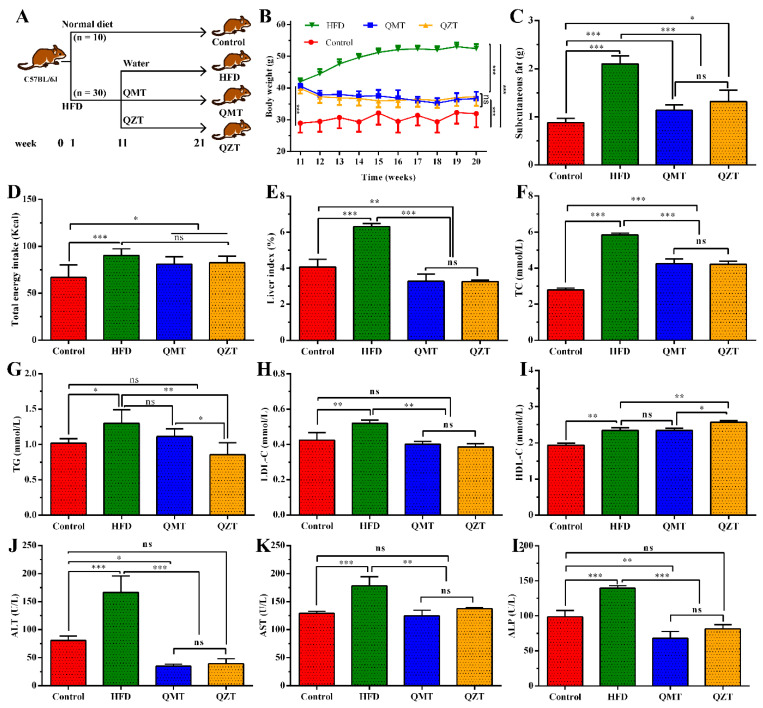
Effects of QMTe and QZTe supplementation on HFD-induced obesity and metabolic disorders. (**A**) Schematic overview of the experimental design. (**B**) The body weight in each group of mice during the 10-week tea intervention. The subcutaneous fat weight (**C**), total energy intake (**D**), liver index (**E**), and the serum levels of TC (**F**), TG (**G**), LDL-C (**H**), HDL-C (**I**), AST (**J**), ALT (**K**), and ALP (**L**) in each group of mice after the 10-week tea intervention.

**Figure 2 foods-11-03210-f002:**
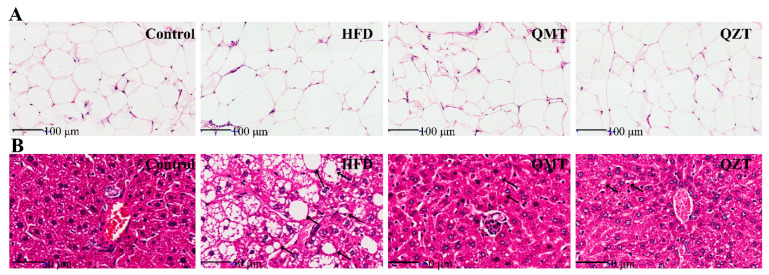
The histopathological change of subcutaneous fat (**A**) and liver (**B**) in each group of mice after 10-week tea intervention.

**Figure 3 foods-11-03210-f003:**
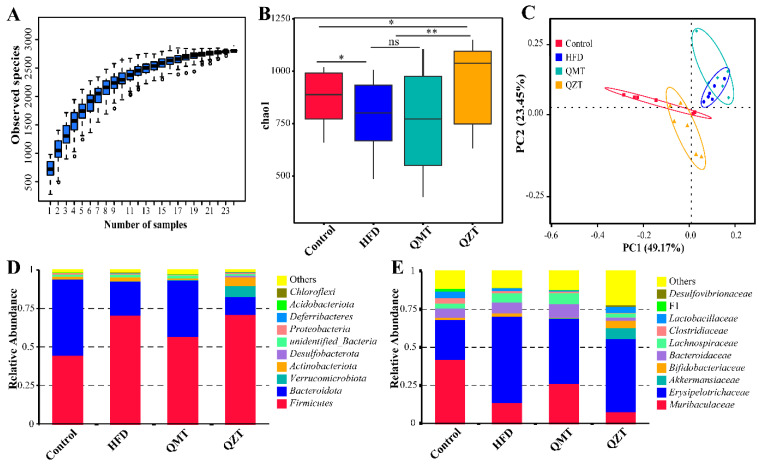
Effects of QMTe and QZTe supplementation on HFD-disrupted gut microbiota composition. (**A**) The species accumulation boxplot. (**B**) chao1 index of gut microbiota in each group of mice. (**C**) PCoA based on OTU relative abundance. Each point represents the composition of gut microbiota in one group. Bacterial taxonomic profiling at the phylum (**D**) and family (**E**) level. F1, *Eubacterium*_*coprostanoligenes*_group.

**Figure 4 foods-11-03210-f004:**
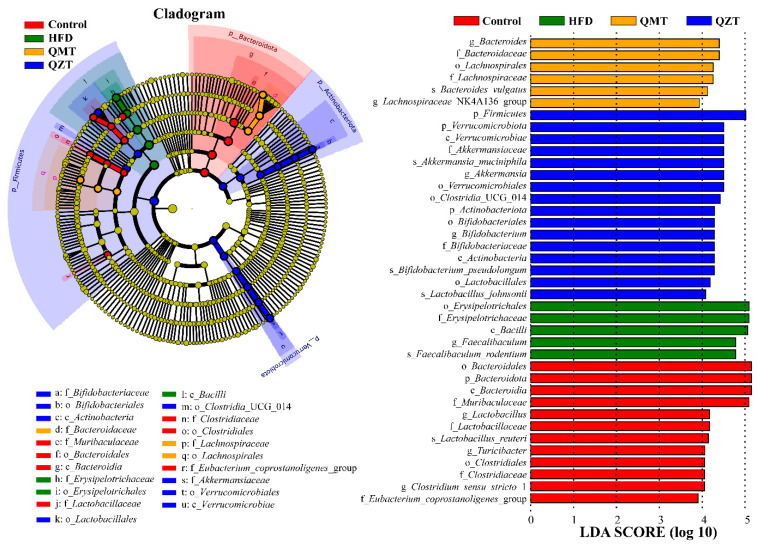
Taxonomic cladogram showed the most differentially abundant taxa in response to HFD, QMTe, and QZTe intervention.

**Figure 5 foods-11-03210-f005:**
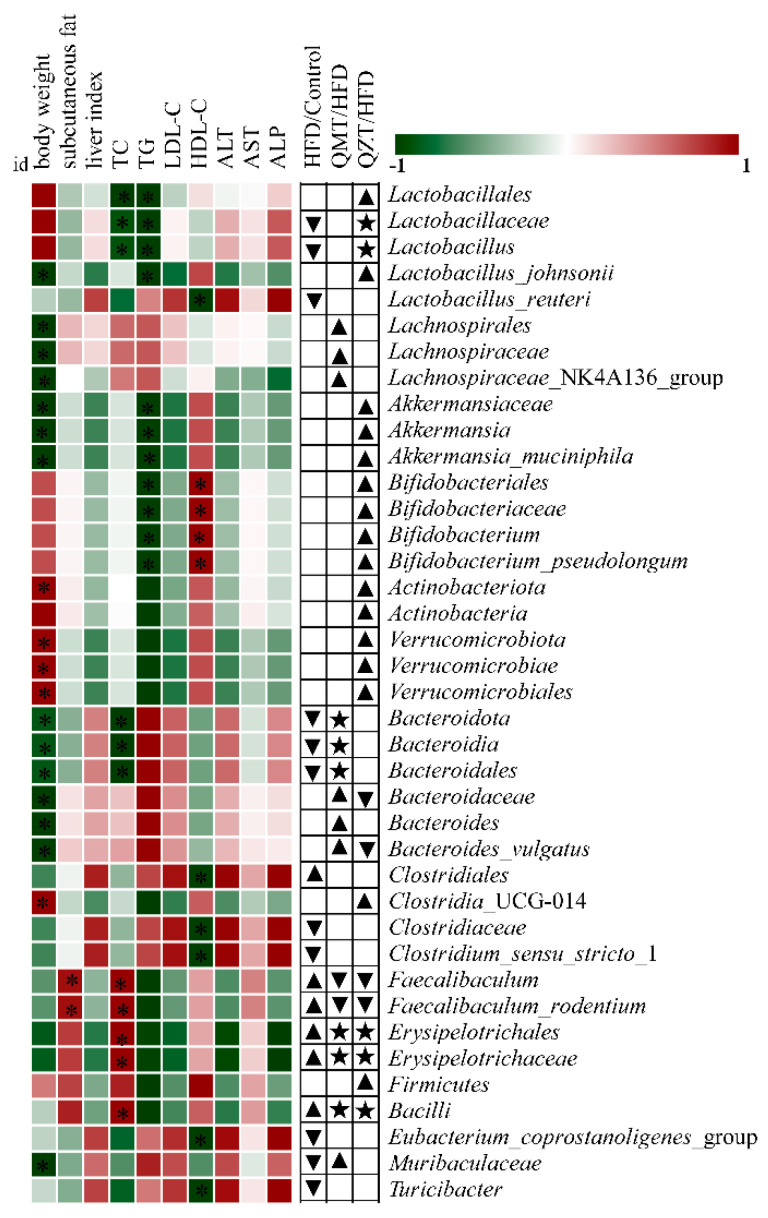
Heatmap of Spearman’s correlation between the most differentially abundant taxa (altered by HFD, QMTe, and QZTe intervention) and obesity-related phenotypes in mice. ▲, significantly more abundant; ▼, significantly less abundant; ★ the microbiota altered significantly by HFD feeding was restored to normal by QMTe/QZTe treatment.

**Figure 6 foods-11-03210-f006:**
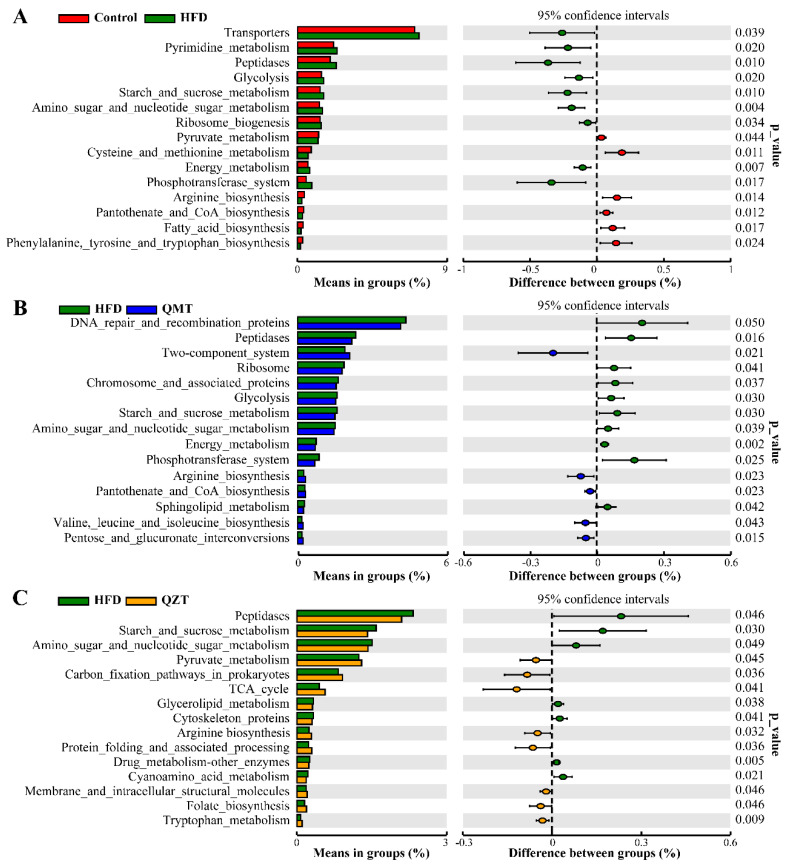
Effects of HFD (**A**), QMTe (**B**), and QZTe (**C**) intervention on microbial community functions predicted by Tax4Fun. TCA_cycle, tricarboxylic acid cycle.

## Data Availability

Data is contained within the article or Appendix A.

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
