# Peer review of "Gut Microbiota Differentially Mediated by Qingmao Tea and Qingzhuan Tea Alleviated High-Fat-Induced Obesity and Associated Metabolic Disorders: The Impact of Microbial Fermentation"

_foods, 2022, doi:10.3390/foods11203210_

Round 1

Reviewer 1 Report

The present study demonstrated the anti-obesity effect of fermented Qingzhuan tea (QZT) and unfermented Qingmao tea (QMT) extracts in high-fat diet-fed mice and correlated with gut microbiota. The parameters like body weight, subcutaneous fat weight, serum levels of TC, TG, LDL-C, HDL-C, AST, ALT, and ALP were studied after 10-week tea intervention. The genomic DNA of the colonic content samples were further studied to explore the gut microbiota diversity.

The manuscript is well designed, executed and written. The English language is also easy going without significant errors.

The manuscript needs the correction of some minor typographical and grammatical errors.

Also, the figures in the manuscript should be revised to have good clarity and resolution.

The figures in the supplementary material can be adjusted in the manuscript.

Reviewer 2 Report

This study explains the difference between fermented QZT and non-fermented QMT in terms of antiobesity effects in HFD-fed mice. Although many studies investigated similar antiobesity parameters as a result of supplementation by different fermented/non-fermented tea types, this is the first study to compare the same type of tea and the antiobesity effects as a result of tea microbial fermentation. In order to better understand the effects of microbial fermentation, a brief literature overview related to this topic would be desirable in the Introduction, e.g. the most important microbial strains/genera involved in the fermentation process, the description of the fermentation conditions and duration, the expected biochemical interactions considering the substrate modulation and generation of microbial metabolites or novel compounds during the fermentation that affect sensorial and health-related properties of the fermented tea, the difference in the composition of the fermented and non-fermented tea in terms of bioactive compounds related to microbial activity…

Section 2.1. A more detailed explanation of the extraction time, vacuum drying and lyophilization should be provided or cited, if published before.

All figures – the resolution must be increased since the data are hardly readable. Figure 1 should be moved to the Results and Discussion section. Explanations of the abbreviations should be added to the figure’s titles.

The explanation of the ACE index should be added.

Some parts of the text (marked in yellow in the attached file) should be rewritten.

Reviewer 3 Report

The manuscript is very interesting, it is well written, except that the general quality of the figures is very low. Especially in figures 1, 3, 4, 6 and S1 and S2, where the texts, axis labels, etc. cannot be seen well and therefore cannot be understood.

Because of this I recommend improving the quality of the figures. For example in Figure 1, the labels for each color used are not distinguished. In the footer of this figure, the meaning of the abbreviations used as TC, TG, LDL-C, HDL-C, AST, ALT, and ALP is not specified. Therefore, it is suggested to take these comments into account to improve it.

Regardless of whether the abbreviations have been mentioned in the text, it is advisable to describe them in the figures where they have been used, so that the figure is self-explanatory and there is no need to search the text of the manuscript where these abbreviations were declared.
